# Fatty Acids as Potent Modulators of Autophagy Activity in White Adipose Tissue

**DOI:** 10.3390/biom13020255

**Published:** 2023-01-30

**Authors:** Karolina Ciesielska, Małgorzata Gajewska

**Affiliations:** Department of Physiological Sciences, Institute of Veterinary Medicine, Warsaw University of Life Sciences (WULS-SGGW), Nowoursynowska 159b, 02-776 Warsaw, Poland

**Keywords:** autophagy, obesity, WAT, fatty acids, ER stress

## Abstract

A high-fat diet is one of the causative factors of obesity. The dietary profile of fatty acids is also an important variable in developing obesity, as saturated fatty acids are more obesogenic than monounsaturated and polyunsaturated fatty acids. Overweight and obesity are inseparably connected with the excess of adipose tissue in the body, characterized by hypertrophy and hyperplasia of fat cells, which increases the risk of developing metabolic syndrome. Changes observed within hypertrophic adipocytes result in elevated oxidative stress, unfolded protein accumulation, and increased endoplasmic reticulum (ER) stress. One of the processes involved in preservation of cellular homeostasis is autophagy, which is defined as an intracellular lysosome-dependent degradation system that serves to recycle available macromolecules and eliminate damaged organelles. In obesity, activation of autophagy is increased and the process appears to be regulated by different types of dietary fatty acids. This review describes the role of autophagy in adipose tissue and summarizes the current understanding of the effects of saturated and unsaturated fatty acids in autophagy modulation in adipocytes.

## 1. Introduction

Obesity is defined as excessive fat accumulation in subcutaneous and abdominal regions of the body, which may lead to serious pathological health conditions, such as coronary disease, diabetes type 2, non-alcoholic fatty liver disease, or arthritis [1]. Individuals with obesity also have a tendency to have pulmonary infections [2], including an increased risk of developing COVID-19 [3], and neurodegenerative diseases such as Alzheimer’s disease [4] and Parkinson’s disease [5]. According to the most recent data, obesity is on the rise and currently affects more than 2 billion people [6]. Numerous crucial factors, such as genetic predisposition, less physical exercise, chronic stress, excessive food intake, sedentary lifestyles, and consumption of highly processed food and alcohol contribute to the development of this condition [7,8]. Moreover, data from epidemiological studies suggest that there is a direct relationship between the amount of dietary fat and the degree of obesity [9]. A high-fat diet (HFD) promotes passive overconsumption, a positive energy balance and weight gain, because high-fat intake does not induce as potent satiety signals as do diets rich in carbohydrates or proteins [10]. Furthermore, fat is more readily absorbed from the intestine and fecal energy loss is much lower with a high dietary fat/carbohydrate ratio [11].

Obesity is accompanied by adipocyte hypertrophy and hyperplasia in white adipose tissue (WAT). The changes observed within fat cells of obese individuals lead to elevated oxidative stress, unfolded protein accumulation, and increased endoplasmic reticulum (ER) stress. These changes disturb adipocytes homeostasis and result in disrupted production of adipokines. Consequently, adipocytes release more proinflammatory cytokines, leading to low-grade inflammation and enhanced infiltration of WAT by macrophages. One of the possible mechanisms involved in preservation of homeostasis in hypertrophic adipocytes is autophagy activation. 

Autophagy is an intracellular recycling process that enables lysosomal degradation of protein aggregates and damaged organelles, and provides essential nutrients in times of energy deprivation or stress conditions. Due to involvement of autophagy in cytoplasm remodeling, it is considered to be important in various stages of development that involve changes in cellular structures during tissue differentiation. Adipogenesis is one of the processes in which autophagy is indispensable, as deletion of autophagy-related genes (*ATGs*) results in impaired WAT development [12,13]. It has been shown that activation of autophagy in adipocytes of obese individuals is elevated; however, it is still not clear if this phenomenon is beneficial or deleterious for WAT functioning, and how it correlates with pathophysiology of metabolic syndrome. 

In this review, we describe in more detail the changes in adipose tissue during obesity and the significance of autophagy in WAT biology. We summarize involvement of several cell signaling pathways in regulation of autophagy in hypertrophic adipocytes, and point out metabolic changes in adipocytes induced by a high-fat diet (HFD). Finally, we discuss the role of different types of fatty acids in the context of altering the signaling pathways that regulate the activity of autophagy in adipocytes.

## 2. Changes in Adipose Tissue during Obesity

Adipose tissue, which was once recognized only as the main site of lipid storage and thus the main source of energy, has been established to be a dynamic endocrine organ synthesizing and secreting active signaling molecules collectively called adipokines, which maintain metabolic homeostasis. Two main types of adipose tissue can be distinguished: white adipose tissue (WAT) and brown adipose tissue (BAT). Both BAT and WAT are present in adults; however, the exact amount of active BAT remains highly variable [14]. BAT is responsible for non-shivering thermogenesis, mediated by the expression of tissue-specific uncoupling protein 1 (UCP1) within the abundant mitochondria that are specific to this type of adipose tissue. Thus, brown adipocytes are capable of rapidly oxidizing their own fat stores and circulating substrates, generating heat and raising metabolic rate [14]. That is why the transition from WAT to BAT is a significant area of interest in obesity research [14,15,16]. Currently, scientific literature also distinguishes beige adipocytes that are derived from the transformation of mature white adipocytes (browning of WAT) or come from de novo differentiation from tissue-resident progenitors. Formation of beige adipocytes can be induced by cold exposure or activation of β-adrenergic receptors. Beige adipocytes share many morphological and functional characteristics with brown adipocytes, such as the presence of multiple small lipid droplets as well as the functions of thermogenesis and energy expenditure. The amount of mitochondria and UCP-1 expression in beige adipose tissue is higher than in WAT but lower than in BAT, reflecting the origin of beige adipocytes perfectly [17]. 

### 2.1. Secretory Profile of Adipose Tissue

The main function of WAT is lipid storage in the form of triacylglycerols (TGs) but, as mentioned earlier, it also actively secretes various adipokines. White adipocytes can secrete hormones such as adiponectin, leptin, resistin, visfatin, as well as cytokines, such as interleukins: 6, 8, 10 (IL-6, IL-8, IL-10), tumor necrosis factor alfa (TNF-α), and monocyte chemoattractant protein-1 (MCP1) [18]. More than any other tissue, healthy adipose tissue has the capacity to drastically expand in response to changes in energy status, and as a result of increased lipid deposition accompanied by lower utilization rate of lipids [19]. This expansion occurs due to increasing size of adipocytes (hypertrophy) and increasing number of adipocytes (hyperplasia) (Figure 1) [20]. 

### 2.2. Stress Induced in WAT

Large, hypertrophic fat cells are typically thought to be less metabolically efficient and are associated with pathophysiological conditions. In adipocytes, enlargement of lipid droplets impairs organelle’s function, e.g., oxidation in the mitochondria, or functions of endoplasmic reticulum (ER). ER is the site of free fatty acids (FFAs) esterification to several classes of lipids, including TGs, cholesteryl esters, and phospholipids. Numerous studies have shown that abnormal intake of fatty acids, especially saturated fatty acids, leads to adipocyte hypertrophy and inhibition of triglyceride lipase [21]. These changes impair adipocytes’ ability to store FFAs in TGs as well as FA beta-oxidation, and encourage the formation of reactive oxygen species (ROS), which limits ER function. In addition, ER homeostasis disruption can impair protein folding process, leading to ER stress and induction of the unfolded protein response (UPR) [22]. Following that, the UPR can trigger the inflammatory response, the anti-oxidative stress response, apoptosis, and other stress response pathways [22]. UPR is mediated by three ER membrane-associated stress-responsive proteins: protein kinase R-like ER kinase (PERK), inositol requiring enzyme 1 (IRE1), and activating transcription factor 6 (ATF6). Activation of these ER-stress sensors leads to stimulation of pro-inflammatory signaling pathways mediated by c-Jun terminal kinase (JNK) and nuclear factor κB (NFκB) transcription factor [23]. Furthermore, ER stress caused by fat accumulation in adipocytes can also lead to activation of adaptive programs, such as ER-associated protein degradation (ERAD) or autophagy [24]. 

### 2.3. Chronic Low-Grade Inflammation

Increased adipocyte size observed in obesity is also linked to enhanced inflammatory response based on the positive correlation between fat cell size and the expression of the inflammatory genes: *NF-κB* and *TNF-α* as well as TNF receptor expression [19]. Adipose tissue in obese people produces higher concentrations of proinflammatory cytokines, including TNF-α, IL-6, IL-8, and simultaneously releases less of the anti-inflammatory IL-10 [25]. It is now commonly recognized that obesity is associated with chronic low-grade inflammation [26,27]. The significance of this inflammatory response in the pathogenesis of obesity-related diseases, such as type 2 diabetes mellitus (T2DM), as well as the potential for anti-inflammatory therapy in obesity, has also been well established by numerous studies [28,29,30]. Furthermore, it is well known that the location of adipose tissue in the body and its amount affect the inflammatory profile of the fat tissue and metabolic consequences [31]. Obese people have different patterns of body fat distribution. Adipose tissue can accumulate in the abdominal cavity, where it is known as visceral adipose tissue (VAT), or in subcutaneous areas (SCATs). In comparison to SCATs, VAT is more cellular, vascular and innervated, and contains more immune cells. It has been demonstrated that abdominal obesity is more likely to be connected with increased production of inflammatory factors, such as high-sensitivity C-reactive protein (hsCRP), and increased infiltration of leukocytes in comparison to subcutaneous adipose tissue [32]. In comparison to SCATs, VAT also shows a lower potential for preadipocyte differentiation and a higher proportion of big adipocytes. Additionally, glucocorticoid and androgen receptors are more abundant in VAT than in SCATs. VAT adipocytes are also more insulin-resistant, more lipolysis-sensitive, and metabolically active compared to SCATs [31]. These factors are crucial for evaluating patients predisposition for metabolic disorders associated with obesity. 

### 2.4. Mitochondrial Dysfunction in WAT

Obesity is also associated with impairment of mitochondrial functions in hypertrophic adipocytes [33]. Free fatty acids that have been properly released as a result of lipolysis undergo the process of beta-oxidation in the mitochondria, which provides cells with energy in the form of ATP. This process is accelerated in obesity due to increased access to fatty acids, resulting in mitochondrial dysfunctions, elevated ROS production, oxidative stress, and inflammatory cytokines secretion. Excess fat accumulation also causes reduced mitochondrial biogenesis and oxidative activity [33]. Dysfunctional mitochondria can be utilized through the process of mitophagy. The removal of mitochondria via autophagy, as well as the decreased mitochondrial biogenesis associated with obesity, could result in far-reaching disruptions in beta-oxidation of fatty acids and metabolic disturbances linked to obesity [33]. High ROS levels also impair preadipocyte proliferation and differentiation, while in mature adipocytes ROS reduces oxygen consumption by blocking fatty acid oxidation, resulting in lipid accumulation [34]. The processes described above play an important role in the pathophysiology of obesity, insulin resistance, type 2 diabetes, and neurodegenerative diseases. ER stress, oxidative stress, and low-grade chronic inflammation may all contribute to increased autophagy activation in hypertrophic adipocytes. 

## 3. Autophagy

Autophagy is derived from the Greek words *autos* (self) and *phagos* (to eat), so the literal meaning is “self-eating”. Autophagy plays a significant role in removing undesirable cellular components, thus maintaining intracellular homeostasis. This process regulates the circulation of cellular proteins and organelles by engaging the controlled breakdown of biochemical compounds with the participation of lysosomes. Three types of autophagy are distinguished, depending on how molecules are delivered for decomposition: chaperone-mediated autophagy (CMA), microautophagy, and macroautophagy. CMA is only involved in protein degradation and relies on chaperone activity of heat shock cognate 70 protein (Hsc70) to identify substrates. Hsc70 can target cargo proteins and deliver them to lysosomes via lysosome-associated membrane protein 2a (LAMP2a) [35]. Microautophagy is the process of digestion of cytosolic components by direct engulfment of cytoplasm into the lysosome. To enclose portions of the cytosol, the lysosomal membrane is randomly invaginated into the autophagic tube during this process [36]. The most well-known type of autophagy is macroautophagy, which has several stages: initiation; formation of an isolating membrane; elongation of the isolating membrane; autophagosome formation; autophagosome-lysosome fusion; cargo degradation. It facilitates the degradation of a wide range of cargo, including proteins, lipids, organelles, and pathogens [37]. Macroautophagy, hereafter referred to as autophagy, at basal level serves as a quality control mechanism in cells, and becomes augmented under stress conditions induced by growth factors deprivation, nutrient deficiency, oxygen deficiency, induction of oxidative stress, or intracellular pathogens [38]. 

### 3.1. Steps of Autophagy Process

Autophagy is activated in response to specific conditions mentioned above, and it is primarily regulated by energy stress sensors including the mammalian target of rapamycin (mTOR, also referred to as the mechanistic target of rapamycin) and adenosine monophosphate-activated protein kinase (AMPK). In mammals, there are six major protein groups involved in autophagosome biogenesis: (1) the unc-51-like kinase (ULK) complex, (2) ATG9-containing vesicles, (3) the autophagy-specific phosphatidylinositol-4,5-bisphosphate 3-kinase (PI3K) complex, (4) the ATG2– WIPI complex (5) the ATG12–ATG5–ATG16L1 complex, and (6) the light chain 3 (LC3) protein lipidation system [39]. AMPK and mTOR initiate the autophagy process by activating ULK complex and PI3K class III complex. In mammals, the ULK complex is composed of ULK1 or ULK2 kinase, Atg13 protein, and FIP200 protein [40,41]. A decrease in mTOR activity causes dephosphorylation of ULK1, ULK2, and thus activation of ULK1 and ULK2, as well as phosphorylation of Atg13 and FIP200 by ULK1 and ULK2 [39]. 

In order to induce formation of autophagosomal membrane, PI3K complex has to be formed. Two types of PI3K complexes are distinguished: PI3K class I and PI3K class III. PI3K class III comprises beclin 1 (BECN1), which is one of the major indicators of autophagy induction in cells, p150, vacuolar protein sorting 34 (VPS34), and ATG14-like protein (ATG14L), or ultraviolet irradiation resistant-associated gene (UVRAG). Beclin1-hVps34-p150-ATG14L protein complex and beclin1-hVps34-p150-UVRAG protein complex are two different versions of PI3K class III complexes [42]. ATG14L and BECN1, subunits of the PI3K complex, and ATG9 are directly phosphorylated by the ULK complex [39,43]. The activation of the ULK kinase complex, which corresponds to the translocation of the ULK complex to a discrete location on the ER marked by ATG9, is the first step in the formation of the phagophore. ATG9 is a multimembrane protein that is localized in the Golgi network, recycling endosomes, cytoplasmic vesicles, and tubules, and it is involved in the formation of autophagosome precursor. The isolation membrane/autophagosome precursor is derived from the ER membranes PI3K class III complex phosphorylates phosphatidylinositol (PI) to produce phosphatidylinositol 3-phosphate (PtdIns3P), which recruits various PtdIns3P-binding proteins that regulate different steps in autophagosome formation. WD-repeat domain phosphoinositide interacting WIPIs (WIPI1–WIPI4) are PtdIns3P-binding proteins in mammals, and they all interact with ATG2 in ATG2-WIPI complex. This complex functions as a membrane tether to the ER, and with lipid transfer activity enables expansion of the autophagosomal membrane (Figure 2). 

The next step is the elongation of the isolating membrane of autophagosome, which involves the inclusion of various proteins. The ubiquitin-like proteins ATG12, ATG5, and ATG16L interact, and thanks to the activity of ATG7 and ATG10 form an ATG12–ATG5–ATG16L1 complex. This complex interacts with WIPI2 and is localized on the autophagosome precursor/isolation membrane [39,43]. The removal of cytosolic ubiquitinated substrates or aggregate-prone proteins is an important function of autophagy in complex organisms. Recent findings suggest that the protein p62 may be involved in cargo recognition. In order to link the ubiquitinated cargos to the autophagy machinery p62 directly binds to LC3 protein [44,45,46]. The LC3 protein is a ubiquitin-like protein that is considered the major marker of autophagic activity due to its association with a fully formed autophagosome. LC3 protein activity is also thought to be the major determinant of autophagosome size [47]. To perform these functions, LC3 must be conjugated with phosphatidylethanolamine (PE) in order to anchor to the isolation membrane/autophagosome precursor. Several proteins participate in this process: ATG4, ATG7, ATG3, and ATG12–ATG5–ATG16L1 complex (Figure 2) [48].

When autophagosome formation is complete, the mature double-membrane autophagosome fuses with lysosomes to degrade its contents. The fusion machinery relies heavily on SNAREs. Before SNARE-mediated fusion can take place, autophagosomes and lysosomes must first move closer together and then become tethered. The outer autophagosomal membrane fuses with the single lysosomal membrane in the first step of autophagosome-lysosome fusion. Full fusion is completed by lysosomal hydrolases degrading the inner autophagosomal membrane and exposing the autophagosome contents to the lysosome lumen [49]. The molecules produced in this process are then released into the cytoplasm, where they serve as substrates for subsequent metabolic processes (Figure 2).

### 3.2. Pathways of Autophagy Regulation 

Autophagy activity in cells is increased in conditions of low energy supply and starvation. It can be regulated by complex networks that include mTOR kinase, which is considered to be the master regulator of autophagy, as well as by mTOR-independent signaling pathways [50]. There are various mTOR-dependent pathways that detect nutrients and energy availability, but this review focuses on the most well-known.

Mammalian target of rapamycin responds to signals induced by hormones and growth factors (e.g., insulin and IGF-I), nutrients (e.g., amino acids), and cellular stress. It can be found in two different complexes in mammals: target of rapamycin complex 1 (mTORC1) and target of rapamycin complex 2 (mTORC2). Despite the fact that these two complexes share the same catalytic subunit: phosphatidylinositol 3-kinase-related kinase (PIKK) and domain-containing mTOR-interacting protein (DEPTOR), they phosphorylate different downstream targets and thus have different cellular functions [51]. The mTORC1 complex is thought to be the main inducer of protein translation, ribosomal biogenesis and inhibitor of autophagy in response to signals such as high energy status, sufficient nutrients availability, presence of metabolic hormones, and growth factors, e.g., insulin and IGF-I. mTORC1 is inhibited by stress mediators, hypoxia, and rapamycin [52]. In adipose tissue, mTORC1 activity is required for proper lipolysis suppression and the maintenance of systemic lipid homeostasis. In the fed state, unrestricted adipose lipolysis interferes with TG clearance and contributes to hyperlipidemia [53]. The multiprotein complex mTORCl is composed of mTOR kinase, a regulatory associated protein of mTOR complex 1 (RAPTOR), MLST8 (or GPL), PRAS40 [54]. The second complex mTORC2 is responsible for cytoskeleton remodeling, controls inflammation, modulates pathways involved in apoptosis and stress responses [52]. This catalytic complex is formed by mTOR kinase, a rapamycin-insensitive companion of mTOR (RICTOR), SINl, MLST8 [54]. Studies have shown that mTORC1 is more sensitive to the effects of rapamycin than mTORC2, though chronic rapamycin administration may also inhibit mTORC2 effects [55].

The RTK/PI3K/AKT/mTORC1 signaling pathway is the most well-known environmentally responsive signal-mediated regulator of autophagy, and it can be induced by growth factors. Proteins and hormones such as insulin activate the class I PI3K complex via tyrosine kinase receptors (RTKs). The class I PI3K complex activates AKT1 kinase, which phosphorylates and inhibits tuberous sclerosis complex (TSC), which has GAP activity towards the RHEB GTPase that promotes mTORC1 kinase activity. TSC is formed of TSC1, TSC2, and TBC1D7 and functions as a negative regulator of mTORC1 signaling. In contrast to other stimuli, amino acids regulate mTORC1 via a TSC complex-independent pathway. This activation process requires the involvement of RAG GTPase [54]. In summary, under nutrient-rich conditions, mTORC1 kinase is active and actively phosphorylates the ULK1/2 complexes and Atg13, thus inhibiting autophagy in cells [56]. Under starvation, cellular stress, or in the presence of rapamycin, mTORC1 is inhibited, resulting in the activation of ULK 1 and ULK2 kinases. As mentioned above, ULK1/2 are responsible for the partial phosphorylation of the Atg13 and FIP 200 proteins, which are required for autophagy activation (Figure 1) [57,58,59]. RAPTOR is an mTORC1 regulatory protein that binds to mTORC1 and provides its function. Lack of RAPTOR inhibits mTORC1 function and activates autophagy under both nutrient-rich and starvation conditions [60]. In vivo studies also demonstrated that RAPTOR deficiency causes autophagic sequestration of lipid droplets, increases lipolysis, and leads to underdevelopment of adipose tissue [61].

Lower energy status in cells also results in an increase in AMPK phosphorylation and activation by LKB1, which inhibits mTORC1 activity by phosphorylating and activating TSC2 or directly binding to RAPTOR [53]. In addition, AMPK leads to ULK1/FIP200-ATG13 and beclin 1 activation, and by all these pathways, serves as a positive regulator of autophagy [42]. AMPK activity increases in response to changes in cellular adenine nucleotide levels, specifically, a decrease in ATP and an increase in AMP and ADP. AMPK is also affected by changes in carbohydrate availability, with glycogen suppressing AMPK activity and low glucose increasing AMPK activity [62]. Furthermore, AMPK activity contributes to autophagy induction during hypoxia, increased cytosolic free Ca^2+^ concentrations, and cytokines that indicate cellular stress [42]. 

Beclin-1 is another important autophagy modulator which is also a component of the class III PI3K complex. In the presence of high energy levels, the anti-apoptotic protein Bcl-2 (B-cell lymphoma/leukemia-2) binds with beclin 1 and prevents autophagy activation [44]. In mice, deletion of the gene encoding beclin 1 (Becn1) resulted in autophagy inhibition, a major reduction in the number of mitochondria, and a significant decrease in the expression of genes involved in fatty acid metabolism in adipose tissue [63]. Moreover, recent studies investigated the role of mTORC2 in the regulation of autophagy. It was demonstrated that mTORC2 phosphorylates beclin-1, GFAP, and voltage-dependent anion channel 1 (VDAC1), thereby preventing autophagy activation [51]. 

### 3.3. Autophagy in Adipocytes 

Autophagy plays a pivotal role in maintaining cellular homeostasis. In the past three decades, selective autophagy has been characterized as a highly regulated and specific degradation pathway for the removal of unwanted cytosolic components and damaged and/or superfluous organelles. Autophagy can also be classified based on the cargo subjected for degradation: pexophagy (selective removal of peroxisomes), nucleophagy, reticulophagy, mitophagy, and lipophagy [37]. The ability of autophagy to contribute to lipo-homeostasis maintenance is especially important in the context of obesity and obesity-related metabolic disorders. It is very likely that the suppression of autophagy and lipophagy as a result of chronic overnutrition promotes lipid accumulation and metabolic compromise. Furthermore, during fasting, lipophagy is acutely activated in the liver in order to rapidly degrade the large lipid bolus delivered from adipose tissue [35].

According to recent findings, enhanced autophagy can have longevity-promoting effects, as determined by genetic experiments conducted in multiple model organisms [64]. Autophagy inactivation caused by knockout of various *ATG*s genes reduces lifespan due to the premature onset of age-related symptoms in several model organisms, underlining the important role of autophagy at early neonatal stages. Despite having only minor abnormalities at birth, the majority of *Atg5*-/- murine neonates died within one day of birth [65]. Conditional knockout of *Atg7* gene by generation of Atg7-deficient mice also caused impairment of autophagosome formation, starvation-induced protein, and organelle degradation [66]. Autophagy is responsible for the continuous turnover of cytoplasmic components. Its absence resulted in the accumulation of abnormal organelles and ubiquitinated proteins. These findings indicate that autophagy is important for the removal of ubiquitin-positive aggregates [66]. Another study revealed that in Becn1F121A/F121A knock-in (KI) mice, the beclin 1/Bcl-2 interaction was disrupted in multiple tissues, which was associated with higher levels of basal autophagic flux [67]. Both male and female KI mice had a significantly longer lifespan than their wild-type littermates [67]. It is well established that autophagy plays a crucial role in energy balance, protein removal, and long life-span, and that activation of this process can be beneficial for cells. As mentioned previously, activation of autophagy is crucial for proper development of adipose tissue. In mouse experimental models and in vitro cultures, deletion of the *Atg5* and *Atg7* genes resulted in underdevelopment of adipose tissue, demonstrating the significance of the autophagy process in adipocyte differentiation and overall lipid balance in the body [12,13]. A recent study has shown that autophagy is required for degradation of inhibitors of adipogenesis, Klf2/3 and C/EBP-β, which is mediated by Atg4 expression. Additionally, it is speculated that autophagy plays a crucial role in cellular remodeling as fat droplets become larger and take up more space within fat cells [68]. Adipocytes isolated from biopsies collected from obese patients showed an increased number of autophagosomes and autolysosomes in the cytoplasm [69]. Increased activity of autophagy in adipocytes of obese individuals may be connected with various stress factors linked with obesity. Excess adipose tissue accumulation is associated with increased ER stress and oxidative stress, hypoxia, and inflammation, all of which may contribute to autophagy activation. However, it is unclear whether the degradation process helps to restore homeostasis or exacerbates cellular stress. 

### 3.4. Obesity and Autophagy Activation Pathways 

As discussed in the previous section, hypertrophic adipocytes are thought to be less metabolically efficient and are exposed to stress conditions connected with impairment of mitochondrial functions and elevated ROS synthesis, ER stress, and inflammation, which are all potent inducers of autophagy. Recent studies have shown that oxidative stress inhibits mTORC1 kinase, leading to autophagy activation in cells [70,71]. In addition, under stress conditions connected with accumulation of misfolded proteins, a phenomenon called ER-phagy (reticulophagy) occurs [72]. ER-phagy is a form of selective autophagy in which endoplasmic reticulum subunits are subjected to degradation. The process involves specific ER-phagy receptors that link ER domains to the autophagy machinery. ER-phagy receptors connect domains of ER membranes with autophagic proteins responsible for autophagosomes formation in cells. These specific receptors bind with members of the ubiquitin-like Atg8 protein family, among which LC3 is the most important. Binding ER-phagy receptors with LC3 occurs via an LC3-interacting region (LIR) motif [72,73]. In some cases, these receptors bind through the FIP200 protein found in the ULK complexes [73]. ER stress activates autophagy by upregulating transcription of autophagy machinery components and modulating microtubule-associated LC3 protein. These effects are linked to upstream unfolded protein response constituents PERK, IRE1, and ATF6, which represent three different signal transduction pathways of UPR. Activation of these pathways upregulates the expressions of proteins that help restore the normal structure of misfolded and unfolded proteins, and may also diminish the ER-stress by elevating autophagy activity in cells, thereby restoring the homeostasis of the endoplasmic reticulum [74]. PERK was reported to upregulate the transcription of numerous autophagy genes and cargo receptors through active transcription factor 4 (ATF4) and CCAAT-enhancer-binding protein homologous protein (CHOP), resulting in an increase in the autophagic flux [75,76]. A recent study on various cell types revealed that ER stress appears to affect LC3 expression predominantly via ATF4-dependent stimulation of LC3 transcription, which makes ATF4 an essential regulator of autophagosome formation, whereas PERK controls the post-sequestration step in the autophagic pathway. IRE1, on the other hand, acts as a negative regulator of autophagy under both ER stress-induced and basal conditions [77].

Under intracellular stress conditions, accumulation of misfolded proteins labeled by the p62 protein activates the ubiquitin-proteasome system (UPS) as well as autophagy, which acts as an integrated quality control system in cells. Through p62 phosphorylation, the proteotoxic stress imposed on by proteasome suppression can trigger autophagy. Interestingly, excessive amounts of p62 delay the delivery of proteasomal substrates to the proteasome even when the proteasomal enzymatic activity is unaffected, so lack of autophagy may disrupt the ubiquitin-proteasome system [78].

Studies on rats fed a high-fat diet (HFD) revealed increased expression of ER-stress-related proteins: glucose-regulated protein 78 (GRP78), the phosphorylated form of IRE1α, and JNK, as well as impairment of insulin receptor signaling [79]. This confirms that HFD causes increased ER stress in white adipose tissue. HFD also elevates the expression of CHOP in adipose tissue, an important component of the unfolded protein response in cells [80]. Increased CHOP expression promotes the inflammatory infiltration of M1 macrophages in WAT, which inhibits the activation of peroxisome proliferator-activated receptor γ (PPARγ) and stimulates the production of pro-inflammatory cytokines, potentially causing insulin resistance. In mice fed HFD, deletion of CHOP enabled PPARγ activity, resulting in M2 macrophage infiltration and increased insulin sensitivity in adipose tissue [80]. Other studies suggest that chronic inflammation caused by the influx of M1 macrophages into adipose tissue and the overexpression of MCP1, IL-1, IL-6, and TNF- by immune cells and adipocytes increases the activity of IκB kinase (IKK), p38 mitogen-activated protein kinase (MAPK), JNK, and protein kinase C (PKC) proteins, which bind to serine residues of the insulin receptor substrate (IRS) and impair tyrosine phosphorylation, resulting in insulin resistance [81]. Furthermore, ectopic fat deposition, mitochondrial dysfunction, lipotoxicity, and ER stress enhance metabolic pathways that may collectively lead to the development of insulin resistance.

ER stress can also lead to apoptotic cell death of adipocytes by increasing the concentration of Ca^2+^ and the level of cellular free fatty acids. Interestingly, this study also demonstrated that adiponectin inhibits ER-stress-induced apoptosis by increasing the level of PPARα mRNA [82]. PPARs are representatives of the nuclear receptor family that act as ligand-inducible transcription factors and play important roles in glucose and lipid metabolism [83]. They occur in three isoforms: PPARα, PPARβ/δ, and PPARγ, and are all also responsible for the inflammatory response. PPARα is found in hepatocytes, enterocytes, and cardiomyocytes, but not in white adipose tissue. The two PPARγ subtypes (PPARγ1 and PPARγ2) are expressed differently in tissues. While PPARγ2 is expressed in adipose tissue, PPARγ1 is highly expressed in brown and white adipose tissue, the large intestine, and immune cells. It was also found in various tissues, including muscles, pancreas, liver, small intestine, and kidneys [84]. Interestingly, higher PPARγ activity in adipose tissue was observed in patients with hypertrophic obesity [85]. PPARs are currently being extensively studied for the possible treatment of various metabolic disorders. PPARα and PPARγ agonists are already being used in clinical settings for the treatment of type 2 diabetes and hyperlipidemia, respectively [83,85]. As a major modulator of adipogenesis, PPAR γ is thought to be effective in modulating obesity. It has been demonstrated that ER-stress regulates the expression of uncoupling protein 1 (UCP1) in beige adipose tissue via PPARγ suppression [86]. It has also been suggested that PPARγ and its agonists can influence the autophagy process in fat calls; however, the exact mechanism has not been fully understood so far [87].

The low-grade inflammation that occurs in obesity can also stimulate autophagy through secretion of proinflammatory cytokines that are able to trigger this intracellular process. In vitro studies on vascular smooth muscle cells demonstrated that TNFα significantly contributed to the pro-autophagic effect via JNK-mediated pathways and inhibition of AKT kinase [88]. It was also demonstrated that TNFα increased the expression of lysosomal/autophagic genes in 3T3-L1 murine adipocytes within 24 h of cells treatment [89]. Moreover, a study on structural changes in murine hypertrophic adipocytes revealed that these fat cells showed ultrastructural and biochemical changes, such as the presence of cholesterol crystals, calcium accumulation, and increased ROS production, which stimulated the inflammasome nucleotide-binding oligomerization domain (NOD)-like receptor (NLR)-3 (NLRP3) and caused adipocyte pyroptosis. Inflammation-induced pyroptosis drew macrophages into adipose tissue, where they aided in the removal of fat cell residues such as lipid material and cholesterol crystals [90].

Signaling pathways involved in autophagy activation in hypertrophic adipocytes are summarized in Figure 3. 

## 4. Fatty Acids Regulation of Autophagy 

The development of obesity is known to be influenced not only by the amount of fat in a diet but also by the composition of fatty acids. Saturated fatty acids (SFAs) have been shown to contribute more to obesity than monounsaturated fatty acids (MUFAs) and polyunsaturated fatty acids (PUFAs) due to the greater propensity for SFAs to be stored in adipose tissue rather than being oxidized [91]. Diets rich in SFAs were shown to induce greater accumulation of body fat and lower satiety than diets enriched in PUFAs [92,93,94]. In addition, excessive SFA consumption causes increased free SFAs in the circulation, as well as elevated expression of genes involved in inflammatory processes in adipose tissue, reduced insulin sensitivity, and increased content of intrahepatic TGs in humans [95,96,97]. High-fat diets containing excess of SFAs lead to hypertrophy of adipocytes and autophagy induction in these cells [98,99]. Studies on an in vitro model of murine 3T3-L1 adipocytes, demonstrated that high concentrations of palmitic acid (0.5–1.0 mM conc. of PA) induced autophagy due to increased ER stress manifested by increased expression of ER stress markers: ATF4, CHOP, p-JNK [98]. In this research, the direct link between autophagy induction and ER stress was confirmed by blocking the ER stress pathway with 4-phenylbutyrate, which caused a significant reduction of LC3-II levels, indicating autophagy suppression. It was hypothesized that increased autophagy activity in hypertrophic adipocytes may serve as a pro-survival mechanism against ER stress and cell death induced by SFAs [98]. Further studies from this research group revealed that unfolded protein response pathways induced by PA treatment of 3T3-L1 adipocytes were attenuated by rapamycin, the potent inducer of autophagy. Rapamycin also prevented nuclear translocation of NFκB, thereby lowering the NFκB-dependent expression of pro-inflammatory cytokines: IL-6 and MCP1 [99]. 

The capacity of fatty acids to accelerate the autophagic flux seems to be correlated with the length of their carbon chains. SFAs containing 15 to 18 carbon atoms as well as *cis*-unsaturated FAs with hydrocarbon chains containing 14 to 20 carbon atoms efficiently stimulate the autophagic flux, whereas shorter and longer fatty acids fail to exert the same effect [100]. It is worth noting that the outcome of lipid overload may be tissue or organ specific. Studies on the outcomes of obesity are also conducted using in vitro models of chondrocytes culture, because obesity strongly contributes to osteoarthritis [101]. Increased levels of free fatty acids in the blood, especially SFAs, may cause elevated expression of matrix metalloproteinases (MMPs) in the extracellular matrix and trigger cartilage degradation. In vitro studies demonstrated that in the presence of SFAs palmitic acid and stearic acid, autophagy activity was increased in chondrocytes, which was manifested by upregulated expression of the key autophagy markers: LC3-II, Atg5 and beclin-1. In contrast, treatment of chondrocytes with lauric acid resulted in decreased autophagy activation, suggesting less stressful effect of this 12-carbon chain SFA in cells [102]. In the case of liver cells, HFD was shown to induce accumulation of autophagosomes in hepatocytes, but impaired the autophagic flux [103]. This effect was caused by long chain SFAs, but not by MUFAs [96]. A comprehensive study on human non-alcoholic steatohepatitis (NASH) and murine liver samples from animals fed HFD also confirmed increased autophagy, manifested by markedly increased expression of the autophagy marker LC3-II in analyzed tissue samples [104]. In this research, human SMMC-7721 and HepG2 hepatoma cells exposed to PA showed higher expression of LC3-II in comparison to cells treated with oleic acid (OA) belonging to MUFAs. Furthermore, a specific knockdown of genes encoding JNK-1 or JNK-2 allowed this research group to conclude that the two JNK isoforms have distinct roles in FFA-mediated autophagy and lipotoxicity. JNK1 promoted PA-induced lipoapoptosis, whereas JNK2 activated pro-survival autophagy and inhibited PA lipotoxicity [104]. Beneficial outcomes of OA treatment was also shown in a study on 3T3-L1-GLUT4-myc adipocytes [105]. Oleic acid improved the activity of insulin-induced signaling pathways by phosphorylating the insulin receptor and inducing AKT phosphorylation [105]. It should be noted that OA significantly increased insulin-stimulated glucose uptake in adipocytes, despite the fact that this monounsaturated fatty acid had no effect on glucose uptake on its own [105]. A study by Niso-Santano and coworkers [106] showed that OA may induce autophagy via a non-canonical pathway in which association of the autophagosome-linked LC3 protein (LC3-II) with the Golgi apparatus is not driven by activation of the multiprotein PI3K class III/VPS34 complex that contains beclin1. Furthermore, autophagic flux stimulated by OA was independent from beclin1 and PI3K class III, and this mechanism was shown to be phylogenetically conserved, occurring in yeast, nematodes, mice, and human cells [106]. The latest report describing an in vivo study on mice fed HFD demonstrating that a long-term exposition to diet rich in unsaturated fatty acids (mainly oleic acid) increased the expression of leptin in WAT and in the blood compared to a diet rich in SFAs palmitic acid and lauric acid [107]. These researchers additionally showed that dietary SFAs weakened the leptin signaling pathways in adipocytes. Nevertheless, diets both rich in unsaturated fatty acids and rich in saturated fatty acids induced hypertrophy in adipose tissue. 

So far, very little is known about possible regulation of autophagy in adipocytes by other types of lipids, such as polyunsaturated fatty acids (PUFAs) or *trans* fatty acids; however, based on the available literature it is possible to assume that the signaling pathways influenced by these fatty acids may also affect autophagy activity in adipose tissue. Beneficial effects of PUFAs on cells of different tissues have been well documented. Our own studies on murine 3T3-L1 adipocytes demonstrated that docosahexaenoic acid (DHA) exerted an anti-inflammatory effect, causing decreased production of pro-inflammatory cytokine IL-6 [108]. Similar effect was shown by De Boer et al. [109], who reported that DHA, and to a lesser extent eicosapentaenoic acid (EPA) decreased MCP1 and IL-6 secretion by adipocytes co-cultured with macrophages. This indicated that n-3 PUFAs may decrease the intensity of pro-inflammatory cross-talk between adipocytes and macrophages. Furthermore, our research group showed that DHA was able to induce lipolysis and increase insulin sensitivity in 3T3-L1 adipocytes [110]. Camargo and co-workers performed a randomized, controlled trial involving 39 obese volunteers with metabolic syndrome, receiving one of four diets for 12 weeks: a high-saturated fatty acid diet (HSFA), a high-monounsaturated fatty acid diet (HMUFA), and two low-fat, high-complex carbohydrate diets supplemented with long-chain n-3 polyunsaturated fatty acids (LFHCC n-3) or placebo (LFHCC) [111]. After the period of dietary intervention, adipose tissue samples were collected from the superficial abdominal subcutaneous adipose tissue. The authors noted a significantly increased expression of autophagy-related *BECN1* and *ATG7* genes after the long-term consumption of HMUFA diet, and an increase in the expression of the apoptosis-related *CASP3* gene after the long-term consumption of the LFHCC and LFHCC n-3 diets. The expression of other analyzed autophagy markers, LC3, LAMP2, and ULK1, tended to increase after the consumption of the LFHCC n-3 diet. The authors concluded that enhanced autophagy may contribute to the maintenance of adipose tissue homeostasis [111]. Another study demonstrated that biologically active derivatives of PUFA may also contribute to autophagy regulation in adipose tissue and the liver [112]. Cytochrome P450 (CYP) epoxygenases catalyze the reaction of epoxides synthesis from PUFAs. Arachidonic acid (AA) is converted into epoxyeicosatrienoic acids (EETs), which act as autocrine or paracrine factors involved in the regulation of inflammation, hyperalgesia, and tissue regeneration, whereas omega-3 PUFAs EPA and DHA are converted into epoxyeicosatetraenoic (EEQs) and epoxydocosapentaenoic (EDPs) acids, respectively. Using fat-1 transgenic mice (capable of producing n-3 fatty acids from the n-6 type, leading to abundant n-3 fatty acids in their organs and tissues, without the need of a dietary n-3 supply), this research group demonstrated that inhibition of soluble epoxide hydrolase (sHE) leads to down-regulation of ER-stress and decreased autophagy in adipose tissue [112]. On the other hand, in livers from obese fat-1 mice, hepatic ER stress was attenuated in the presence of sEH inhibitor but autophagy was restored. A similar result was noted by a different research group, who analyzed the effect of different fatty acids in McA hepatic cell line and noted that DHA stimulated autophagy by increasing the ratio of LC3-II to LC3-I in the absence and presence of lysosomal inhibitors [113]. In the case of adipose tissue, inhibition of autophagic function in obesity may be regarded as beneficial in terms of lipid homeostasis and metabolic control, as it is related to reduced fat mass and improved insulin sensitivity [114]. The studies described above indicate that PUFAs may be involved in autophagy regulation directly, but also indirectly, via their small bioactive lipid mediators.

*Trans*-unsaturated fatty acids, which contain one or more unconjugated double bond in the *trans* configuration, are recognized as a group of FA with known negative effects on human health. Most *trans*-unsaturated FAs are generated through partial hydrogenation of vegetable oils rich in PUFAs, which is an intended industrial process. Many countries have passed laws restricting the use of industrial *trans* fats (triacylglycerols that are rich in *trans*-unsaturated FA) in food products because there is a recognized link between eating these fats and developing cardiovascular disease [115,116]. A growing body of evidence points to a significant role of *trans*-unconjugated FAs in development of insulin resistance and nonalcoholic fatty liver disease [117,118]. However, *trans*-unsaturated FAs cannot be completely removed from human diet, due to their innate presence in ruminant animal meat and dairy products, with *trans*-vaccenic acid being the predominant naturally occurring *trans* FA [119]. According to in vitro and in vivo studies, industrial *trans* fats increase inflammatory cytokine production and ER stress in adipose tissue [120]. In mice, HFDs rich in industrial *trans* fats promoted liver fat storage over fat storage in other localizations [121]. An in vitro study on cardiac myofibroblasts demonstrated that 24 h incubation with *trans* fatty acids *trans*-vaccenic and *trans*-elaidic acid resulted in autophagy induction, which was connected with the stress induced by these fatty acids in myofibroblasts that culminated in premature apoptosis [122]. Interestingly, a study by Sauvat and coworkers [123] performed on the U2OS cell line of human bone osteosarcoma with epithelial morphology showed that and *trans*-unsaturated FAs were less active in eliciting cellular stress than *cis*-unsaturated FAs, as well as saturated FAs. These *trans*-unsaturated FAs, mainly elaidic acid and linoelaidic acid, did not stimulate autophagic response in U2OS cells [123]. The authors observed that elaidic acid (present in processed food) was not able to prevent autophagy induced by starvation or rapamycin (and rapalogs), but this *trans*-unsaturated FA inhibited autophagy induced in the presence of SFA palmitic acid. The effect seemed specific, because elaidic acid only prevented autophagy and ER stress induced by PA, but not by the unsaturated OA. Furthermore, the observation was independent of the sequence of administration of tested fatty acids. The authors hypothesized that the autophagy-inhibitory effect of *trans*-unsaturated FAs could result from their capacity to abolish cytoprotective stress responses induced by SFAs [123]. 

Not all fatty acids containing double bonds in *trans* configuration are linked with adverse effects on human health. Conjugated linoleic acid (CLA) is an octadecadienoic acid characterized by two conjugated double bonds. Among 28 possible isomers of CLA, *cis*-9, *trans*-11 (c9, t11 CLA) isomer is the most abundant in food, contributing to about 90% of CLA in the diet [124]. Ruminant meat and dairy products are the major dietary sources of c9, t11-CLA, whereas partially hydrogenated oils such as margarines are the main sources of *trans*-10, *cis*-12 CLA isomer (t10, c12 CLA) [125]. These two CLA isomers are characterized with the greatest biological activity [126]. c9, t11 CLA has been shown to exert anticarcinogenic and anti-inflammatory properties [127,128]. On the other hand, t10-c12 CLA has antiobesity effects, causing reduced body fat and increased lean muscle mass in several animal models [129,130]. Yeganeh and coworkers [131] examined the effects of c9, t11 CLA and t10, c12 CLA isomers as well as caloric restriction in obese *db/db* mice and lean C57BL/6J mice, investigating several systemic factors such as body composition or glucose control, and characterizing adipose tissue functions by analyzing cell turnover (presence of stem cells and preadipocytes) as well as apoptosis and autophagy in adipocytes. The authors observed that t10, c12 CLA isomer was the main inducer of weight loss, and its effect was independent of food intake [131]. The weight loss resulted primarily from fat loss and increased number of cells with nuclear localization of the beige cell marker Tbx-1, whereas caloric restriction induced different mechanisms, based on reduced lean and fat mass and blocking adipocytes differentiation. Interestingly, the study also showed that significant fat loss obtained by administration of t10, c12 CLA isomer was not connected with altered activity of autophagy in adipocytes (assessed by the expression of LC3-II autophagy marker), thus the effect of CLA was unlikely due to dysregulated autophagy [131]. Corresponding observations regarding autophagy regulation by CLA were obtained in a study using an in vitro model of bovine mammary epithelial cells (bMECs) challenged with lipopolysaccharide (LPS), which mimicked the state of mastitis in bovine mammary gland [132]. LPS induced oxidative stress and inflammatory response of bMECs manifested by increased concentration of ROS and TBARS, decreased concentration of GSH expression, and elevated expression of interleukins: IL-1β, IL-6, IL-8. In addition, LPS administration upregulated the levels of autophagic proteins: LC3B, beclin1, and Atg5. However, when bMECs were pretreated with c9, t11 CLA isomer, the accumulation of ROS was diminished and the expression of autophagy markers was lower despite the LPS challenge. The authors also observed that c9, t11 CLA isomer promoted the milk fat synthesis-related gene expression and lipid droplet formation in BMECs [132]. Results of the aforementioned studies showing decreased autophagy activity in the presence of CLA point at the beneficial properties of this PUFA, which diminishes the stress response of cells, and therefore ameliorates the activity of autophagy induced as a survival mechanism.

A summary of recent findings regarding the effects of different types of fatty acids on autophagy in adipocytes is presented in Table 1. 

## 5. Conclusions and Future Directions

Autophagy is a cytoprotective catabolic process induced by various stimuli, such as nutrient deprivation or other changes in extracellular and intracellular conditions that induce cell damage. In adipose tissue, autophagy plays a major role in the maintenance of homeostasis, as it is involved in adipocytes differentiation, and becomes enhanced in hypertrophic fat cells in obesity when adipocytes are exposed to increased oxidative stress and ER stress, which potentially lead to WAT dysfunction. High-fat diet has been proven to accelerate autophagy activity in adipose tissue; however, the composition of dietary fatty acids appears to influence the direction of autophagy modulation. SFAs, which are the most abundant natural fatty acids present in high-fat diets, induce autophagy, which seems to have a direct correlation with activation of the diabetogenic stress kinase JNK1 as well as increased ER stress observed in hypertrophic adipocytes. On the other hand, PUFAs exert an opposite effect, being able to alleviate SFA-induced mitochondrial dysfunctions, diminish oxidative stress, and modulate the inflammatory response of the adipose tissue. Still, our knowledge about signaling pathways modulated by different types of fatty acids in adipocytes is insufficient. New insights into fatty acids function in relation to autophagy modulation may provide useful tools in planning diet modifications during obesity treatment.

## Figures and Tables

**Figure 1 biomolecules-13-00255-f001:**
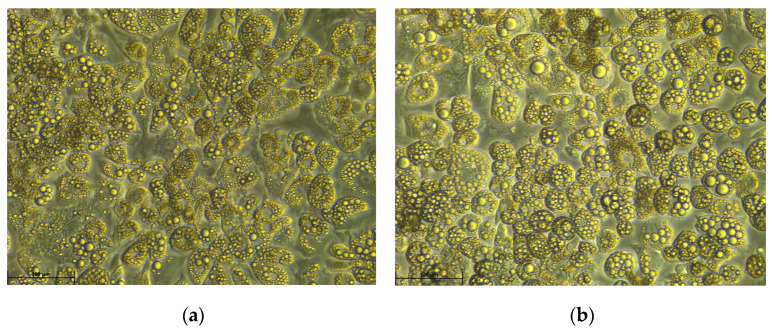
Phase-contrast microscopy images of control (**a**) and hypertrophic (**b**) 3T3-L1 cells on day 12 of adipogenic differentiation (100× magnification). Hypertrophy was induced by addition of palmitic acid (PA, 0.5 mM) to DMEM (+10% FBS) culture medium. 3T3-L1 cells were first subjected to differentiation in DMEM (+10% FBS) culture medium supplemented with 3-isobutyl-1-methylxanthine (IBMX, 0.5 mM), dexamethasone (DEX, 1 μM), insulin (0.5 μM), and rosiglitazone (2 μM) for 3 days, followed by 3 day culture in DMEM (+10% FBS) culture medium comprising insulin (0.5 μM) and rosiglitazone (2 μM). PA (0.5 mM) was administered to 3T3-L1 adipocytes from day 8 to day 12 of differentiation. Enlarged lipid droplets in adipocytes presented on image (**b**) are a characteristic feature of hypertrophy.

**Figure 2 biomolecules-13-00255-f002:**
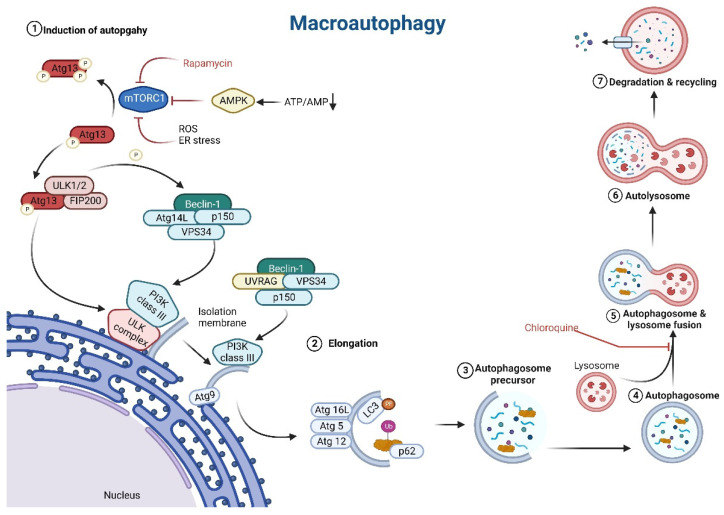
Steps of autophagy: (**1**) Induction of autophagy: decreasing ATP to AMP ratio activate AMPK; AMPK inhibits mTORC1 which phosphorylates ATG13 protein; Active form of ATG13 form a complex with FIP200 and ULK1/2 protein; ULK complex activates PI3K class III complex composed of BECN1, p150, VPS34 and ATG14L, or UVRAG; ULK complex also directly phosphorylates ATG9; PI3K complex III with ATG9 contribute to autophagosome membrane formation. (**2**) Elongation: ATG12–ATG5–ATG16L1 complex contribute to autophagosome membrane elongation, active LC3 conjugated with phosphatidylethanolamine (PE) allows p62 protein to bind ubiquitinated cargos. (**3**–**5**) After enclosure of the autophagosomal membrane surrounding the cytosolic cargo (macromolecules and/or damaged organelles), the mature double-membrane autophagosome fuses with lysosomes to degrade the carried cargo. (**6**) Full fusion is completed by lysosomal hydrolases degrading the inner autophagosomal membrane and exposing the autophagosome contents to the lysosome lumen. (**7**) The simple molecules produced during lysosomal degradation are released into the cytoplasm, where they serve as substrates for subsequent metabolic processes.

**Figure 3 biomolecules-13-00255-f003:**
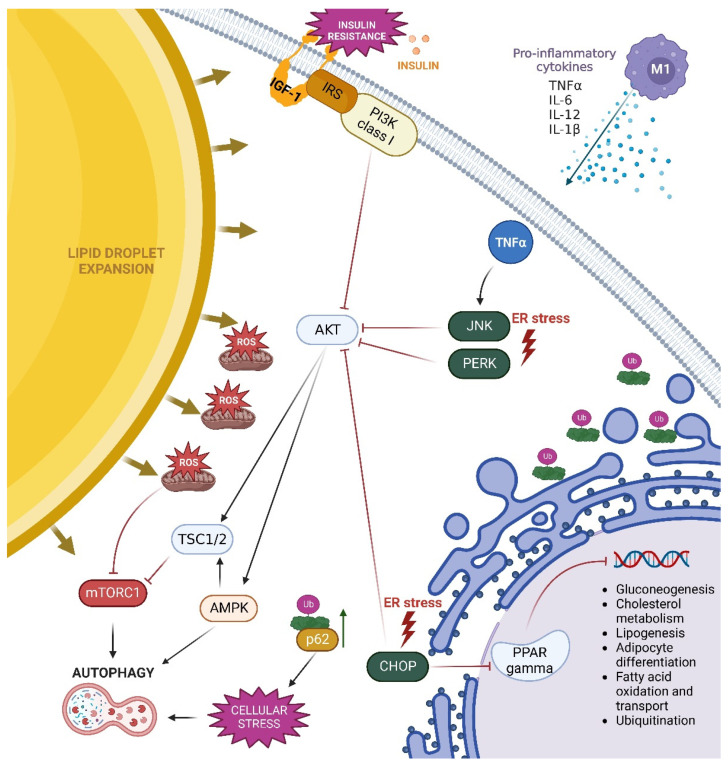
Pathways of autophagy activation in hypertrophic adipocytes in obesity. Enlargement of lipid droplets in hypertrophic adipocytes impairs mitochondrial oxidation and endoplasmic reticulum (ER) functions. Increased fatty acid content accelerates the process of beta-oxidation in obesity, resulting in mitochondrial dysfunction, elevated ROS production, oxidative stress, inflammatory cytokine secretion, and compromised ER function. These changes activate the ER-stress sensors PERK, JNK, and CHOP, which enhance autophagy by inhibiting AKT kinase. Additionally, increased TNF-α concentration enhances JNK-mediated pathway and inhibits AKT activity. Increased ROS production has an immediate effect on mTORC1, limiting its function and activating the autophagy process. Insulin resistance occurring in obesity also leads to diminished activation of the PI3K/AKT, which in turn promotes activation of TSC1/2 and AMPK. AMPK directly promotes autophagy, whereas TSC1/2 is involved in autophagy induction by inhibiting the mTORC1 complex. Furthermore, increased CHOP expression promotes PPAR inhibition. Active PPAR is involved in gluconeogenesis, cholesterol metabolism, lipogenesis, adipocyte differentiation, fatty acid oxidation and transport, and ubiquitination. In obesity, all of these processes are disrupted.

**Table 1 biomolecules-13-00255-t001:** Summarized effects of fatty acids on autophagy in adipocytes.

Type of Fatty Acid	PA Saturated Fatty Acid	OA Monounsaturated Fatty Acid	DHA, EPA Polyunsaturated, Omega-3 Fatty Acid	CLA Polyunsaturated, Conjugated Omega-6 Fatty Acid
Effect in the cell	Elevated ER stress markers: ATF4, CHOP, p-JNK and induction of UPR; increased ROS production; weakened leptin signaling pathways in adipocytes; promotion of insulin resistance.	Improved activity of insulin-induced signaling pathways; increased leptin expression in WAT; inhibited cellular ROS production.	Anti-inflammatory effect (reduced production of IL-6); induced lipolysis increased insulin sensitivity in adipocytes; reduced cellular ROS production; PPARγ agonists.	Anti-inflammatory properties; anti-obesogenic effects (lowering body fat and increasing lean muscle); diminished ROS production; PPAR agonists.
Effect on autophagy	Increased autophagy induction, but impaired autophagic flux.	Increased autophagy induction and increased autophagic flux.	Increased expression of autophagy markers.	Autophagy on basal level.
Conclusion	Increased autophagy activity may serve as a pro-survival mechanism, but in the case of PA, autophagy is also impaired, which can exacerbate adipocytes dysfunction.	Increased autophagy activity may serve as a pro-survival mechanism against cellular stress.	Increased autophagy activity may serve as a pro-survival mechanism against cellular stress.	No documented direct influence on autophagy, improvement of secretory profile and metabolic level.

## Data Availability

Not applicable.

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
