# Peer review of "Fatty Acids as Potent Modulators of Autophagy Activity in White Adipose Tissue"

_biomolecules, 2023, doi:10.3390/biom13020255_

Round 1

Reviewer 1 Report

1. The title can be written more briefly: 

for example; fatty acids as potent modulators of autophagy activity in white adipose tissue

2. Section 2 "Changes in adipose tissue during obesity" can be divided into subsections for better understanding

Author Response

We would like to thank the Reviewer for evaluation of our manuscript entitled High fat diet and different types of fatty acids as potent modulators of autophagy activity in white adipose tissue. Your thoughtful and thorough remarks and suggestions helped us improve the quality of our manuscript.

We followed your suggestion regarding the title and changed it according to your guidelines. The new title is “Fatty acids as potent modulators of autophagy activity in white adipose tissue”.

In addition, Section 2 "Changes in adipose tissue during obesity" has been divided into four subsections for better clarity:

2.1 Secretory profile of adipose tissue

2.2 Stress induced in WAT

2.3 Chronic low-grade inflammation

2.4 Mitochondrial dysfunction in WAT  

We also corrected the style and punctuation in the manuscript using suggestions presented by Reviewer 2 and Reviewer 3. All changes made to the manuscript have been marked up using the “Track Changes” function in the MS Word file. 

Once again we would like to thank you for your valuable contribution and for taking the time and effort to read and review our manuscript.

Yours Sincerely,

Katarzyna Ciesielska and Małgorzata Gajewska

Reviewer 2 Report

The presented review is well-written and organized, and the topic is significant, considering human fat consumption and its role in the development of obesity. I have only some minor comments/suggestions: 

1. I need to see in the text the support for the second sentence of the abstract (lanes 8 and 9). If fatty acids have a similar amount of calories per gram, how do you support that some FAs are more obesogenic than others? Same idea in lanes 33 and 34; your affirmation considers fat responsible when calories and other macronutrients such as carbohydrates are totally ignored.

2. In general, you should be very cautious in the use of terms as 'increase autophagy' (or 'inhibited autophagy' at the end) because some studies presented only show the increase in markers of autophagy and not autophagy flux or activity. For example, the increase in LC3 II can mean an increase or decrease in the autophagic process. As for Table 2, what do you mean by 'Increased activity of effective autophagy"?

Note: This point is even more critical if you consider the affirmation you made in lanes 536-538 about OA modulation of autophagy flux.

3. Lane 635, the best word is 'food intake,' not feed intake.

4. I think the paragraph between lanes 545 and 588 should focus better on autophagy.

Please consider all comments/suggestions to modify, if necessary, tables and illustrations.

Author Response

Response to Reviewer 2

We would like to thank the Reviewer for evaluation of our manuscript entitled High fat diet and different types of fatty acids as potent modulators of autophagy activity in white adipose tissue. Your thoughtful and thorough remarks and suggestions helped us improve the quality of our manuscript. Detailed response to your remarks is presented below:

Comment 1: I need to see in the text the support for the second sentence of the abstract (lanes 8 and 9). If fatty acids have a similar amount of calories per gram, how do you support that some FAs are more obesogenic than others? Same idea in lanes 33 and 34; your affirmation considers fat responsible when calories and other macronutrients such as carbohydrates are totally ignored.

Response: This was a very important remarked. We included additional information about high-fat diet (HFD) in relation to obesity to support the first two sentences of the abstract: A high-fat diet is one of the major causative factors of obesity. The dietary profile of fatty acids is also an important variable in developing obesity, as saturated fatty acids are more obesogenic than monounsaturated and polyunsaturated fatty acids.  

First, we explained in lines 35-41 the obesogenic effect of HFD, by adding information about the crucial characteristics of this type of diet, which promotes passive overconsumption, a positive energy balance and weight gain, due to the fact that high-fat intake does not induce as potent satiety signals as do diets rich in carbohydrates or proteins. Furthermore, fat is more readily absorbed from the intestine, and fecal energy loss is much lower with a high dietary fat/carbohydrate ratio. Appropriate articles were cited to support these findings [Doucet E, Tremblay A. 1997; Astrup A. 2005].

The explanation of highly obesogenic nature of saturated fatty acids(SFA) in comparison to unsaturated fatty acids was provided at the beginning of chapter 4: Fatty acids regulation of autophagy (lines 511-517). We included information about propensity for SFA to be stored in adipose tissue rather than being oxidized. In addition, SFA induce greater accumulation of body fat and lower satiety than diets enriched in PUFA, and excessive SFA consumption causes increased free SFA in the circulation, as well as elevated expression of genes involved in inflammatory processes in adipose tissue, reduced insulin sensitivity and increased content of intrahepatic TGs in humans.  Appropriate articles were also included in the references [Storlien LH, et al., 1998; Lawton CL, et al. 2000; Moussavi N, et al. 2008; Phillips CM, et al. 2012; van Dijk SJ, et al. 2009; Rosqvist F, et al. 2014; Melo HM, et al. 2019].

In addition, we would like to underline that we did not include information in the manuscript about the effect of high-carbohydrate diet in obesity for better clarity of the topic described. Obesity is induced by many factors and the presence of other macronutrients, such as carbohydrates influences the amount of calories consumed. Nevertheless, we focussed on the role of dietary fat, which is stored in adipose tissue when consumed in excessive amounts, directly influencing adipocytes functions and metabolism. 

Comment 2: In general, you should be very cautious in the use of terms as 'increase autophagy' (or 'inhibited autophagy' at the end) because some studies presented only show the increase in markers of autophagy and not autophagy flux or activity. For example, the increase in LC3 II can mean an increase or decrease in the autophagic process. As for Table 2, what do you mean by 'Increased activity of effective autophagy"? Note: This point is even more critical if you consider the affirmation you made in lanes 536-538 about OA modulation of autophagy flux.

Response: We would like to thank the Reviewer for this important comment, with which we entirely agree. We made necessary corrections in the text in order to clearly state whether the studies described in chapter 4: Fatty acids regulation of autophagy determined changes in the expression of autophagy markers, or provided evidence for increased/decreased autophagy flux in the presence of specific fatty acids. We also corrected Table 2, deleting the incorrect term: “Increased activity of effective autophagy” and replacing it with “Increased autophagy induction and autophagic flux”.

Comment 3: Line 635, the best word is 'food intake,' not feed intake

Response: Thank you, this mistake has been corrected (line 699 in the new version of the manuscript).

Comment 4: I think the paragraph between lanes 545 and 588 should focus better on autophagy.

Response: Thank you for this comment. The data on the effect of PUFA on autophagy activity in adipose tissue are very limited. However, we have rewritten this paragraph, which now focusses more on autophagy than on the general effect of PUFA. Some additional studies have been described and the cited articles have been added to the list of references [Camargo A, et al. 2017; López-Vicario C, et al. 2015; Caviglia JM, et al. 2011; Singh R, et al. 2009]. The corrected paragraph is presented in lines 618-651 of the new version of the manuscript.

In addition, we would like to inform that the title of the manuscript has also changed, following the suggestion of Reviewer 1.

The new title is “Fatty acids as potent modulators of autophagy activity in white adipose tissue”.

All changes made to the manuscript have been marked up using the “Track Changes” function in the MS Word file. 

Once again we would like to thank you for your valuable contribution and for taking the time and effort to read and review our manuscript.

Yours sincerely,

Katarzyna Ciesielska and Małgorzata Gajewska

Reviewer 3 Report

The manuscript is a review of the modulation processes of autophagy in white adipose tissue, particularly in obesity. In my opinion, the review has interest but I believe thast it needs a revision of style and punctuation that will improve clarity. Some suggested changes are listed below:

- In line 44, in the introduction, the authors begin to talk about macroautophagy without having previously explained the three types of autophagy, which are explained later.

- In line 47, important on various stages - important in various stages

- In line 48, tissues differentiation - tissue differentiation

- In line 50, the sentence lacks a colon after the references 10, 11

- In line 54, I think the correct phrase has to be: in more detail

- In line 55, include the definite article: We summarize the involvement

- In lines 74-75, I think it is necessary to explain what beige adipocytes are

- In lines 76-77, the position of the comma has to be: but, as mentioned earlier,

- In lines 83-84, include the definite article: due to the increasing size of adipocytes (hypertrophy) and the increasing number of adipocytes

- In lines 129-130, is more cellular, vascular, and innervated,

- In line 133,  use the abbreviation: in comparison to SCAT

- In line 469, the figure legend lacks a colon.

- Table 2 is difficult to read, I would remove the hyphens. Decide if you use increase or increased.

Author Response

Response to Reviewer 3

We would like to thank the Reviewer for evaluation of our manuscript entitled High fat diet and different types of fatty acids as potent modulators of autophagy activity in white adipose tissue. Your thoughtful and thorough remarks and suggestions helped us improve the quality of our manuscript. Detailed response to your remarks is presented below:

General response to all suggested changes of style and punctuation: We would like to sincerely thank the Reviewer for such detailed revision of our manuscript and for pointing out many mistakes in the punctuation as well as stylistic errors. All of the listed mistakes have been corrected and the changes made to the manuscript have been marked up using the “Track Changes” function in the MS Word file, based on the instruction of the Biomolecules journal editorial office. 

Comment regarding line 44: in the introduction, the authors begin to talk about macroautophagy without having previously explained the three types of autophagy, which are explained later.

Response: Thank you for this comment – we replaced the word macroautophagy with autophagy in chapter 1: Introduction (lines 51-62 in the new version of the manuscript).

Comment regarding line 55: include the definite article: We summarize the involvement

Response: This is the only suggestion which was unclear for us, therefore was not included in the corrections. The Reviewer refers to the last paragraph of chapter 1: Introduction, in which we presented a short summary of the topics which are described in the main text of this review article, namely: (1) involvement of several cell signaling pathways in regulation of autophagy in hypertrophic adipocytes, (2) metabolic changes in adipocytes induced by high fat diet (HFD), (3) the role of different types of fatty acids in the context of altering the signaling pathways that regulate the activity of autophagy in adipocytes. Therefore this paragraph does not contain any citation.

Comment regarding line 74-75: I think it is necessary to explain what beige adipocytes are

Response: Thank you for this suggestion. Information about beige adipocytes have been added (lines 83-87 of the new version of the manuscript).

Comment regarding line 83-84: include the definite article: due to the increasing size of adipocytes (hypertrophy) and the increasing number of adipocytes

Response: Thank you for this suggestion. A correct citation was added to the body text (line 97 of the new version of the manuscript) and to the list of references (article number 20).

Comment: Table 2 is difficult to read, I would remove the hyphens. Decide if you use increase or increased

Response: Thank you for this comment. The hyphens have been removed from the body text of
table 2 to improve its clarity. We corrected “increase” to “increased”. In addition, following suggestions of Reviewer 2 we replaced the term: “Increased activity of effective autophagy” with “Increased autophagy induction and autophagic flux”.

Once again we would like to thank you for your valuable contribution and for taking the time and effort to read and review our manuscript.

Yours sincerely,

Katarzyna Ciesielska and Małgorzata Gajewska